# Detraining and Retraining Effects from a Multicomponent Training Program on the Functional Capacity and Health Profile of Physically Active Prehypertensive Older Women

**DOI:** 10.3390/healthcare12020271

**Published:** 2024-01-21

**Authors:** Luís Leitão, Yuri Campos, Hugo Louro, Ana Cristina Corrêa Figueira, Teresa Figueiredo, Ana Pereira, Ana Conceição, Daniel A. Marinho, Henrique P. Neiva

**Affiliations:** 1Sciences and Technology Department, Superior School of Education of Polytechnic Institute of Setubal, 2910-761 Setúbal, Portugal; ana.figueira@ese.ips.pt (A.C.C.F.); teresa.figueiredo@ese.ips.pt (T.F.); ana.fatima.pereira@ese.ips.pt (A.P.); 2Life Quality Research Centre, 2040-413 Rio Maior, Portugal; 3Department of Sport Sciences, University of Beira Interior, 6201-001 Covilhã, Portugal; marinho.d@gmail.com (D.A.M.); hpn@ubi.pt (H.P.N.); 4Post Graduate Program in Physical Education, Federal University of Juiz de Fora, Juiz de Fora 36036-900, Brazil; reiclauy@hotmail.com; 5Study Group and Research in Neuromuscular Responses, Federal University of Lavras, Lavras 37200-900, Brazil; 6Department of Sport Sciences, Sport Sciences School of Rio Maior, Polytechnic Institute of Santarém, 2040-413 Santarém, Portugal; hlouro@esdrm.ipsantarem.pt (H.L.); anaconceicao@esdrm.ipsantarem.pt (A.C.); 7Research Center in Sports Sciences, Health Sciences and Human Development (CIDESD), 6201-001 Covilhã, Portugal

**Keywords:** older women, detraining, retraining, hypertension, multicomponent training

## Abstract

Background: Resuming a physical exercise program after a period of cessation is common in older women. Monitoring the responses during this detraining (DT) and retraining (RT) may allow us to analyze how the body reacts to an increase and a reduction in physical inactivity. Therefore, we conducted a follow-up training, DT, and RT in prehypertensive older women to analyze the response to these periods. Methods: Twenty-three prehypertensive older women (EG; 68.3 ± 2.8 years; 1.61 ± 0.44 m) performed 36 weeks of the multicomponent training program (MTP) followed by twelve weeks of DT plus eight weeks of RT. Fifteen prehypertensive older women (CG; 66.3 ± 3.2 years; 1.59 ± 0.37 m) maintained their normal routine. Functional capacity (FC), lipid, and hemodynamic profile were assessed before, during 24 and 36 weeks of the MTP, after 4 and 12 weeks of DT, and after 8 weeks of RT. Results: After 24 weeks of the MTP, only SBP did not improve. Four weeks of DT did not affect lower body strength (30-CS), TC, or GL. Eight weeks of RT improved BP (SBP: −2.52%; ES: 0.36; *p* < 0.00; DBP: −1.45%; ES: 0.44; *p* < 0.02), handgrip strength (3.77%; ES: 0.51; *p* < 0.00), and 30-CS (3.17%; ES: 0.38; *p* < 0.04) compared with 36 weeks of the MTP. Conclusions: Eight weeks of RT allowed patients to recover the benefits lost with detraining, which after only four weeks affected them negatively, and the systematic practice of exercise contributed to greater regulation of BP since 24 weeks of the MTP proved not to be enough to promote positive effects of SBP.

## 1. Introduction

Prehypertension can be seen as a phase of gradual changes from optimum blood pressure (BP) to hypertension [1], with threshold values of 120 to 139 mmHg systolic (SBP) or 80 to 89 mmHg diastolic (DBP) [2]. Individuals with an SBP of 120–140 mmHg fall into a “danger zone” [3] compared to normotensive people, who have an increased risk of developing hypertension and cardiovascular disease [4]. Hypertension remains the leading cause of death worldwide (>10.4 million deaths per year). It may cause damage to the heart through hardening of the arteries, reducing blood flow and oxygen perfusion to the heart muscle and other tissues [5]. A sedentary lifestyle is considered the main modifiable risk factor for hypertension development [6]. On the other hand, physical exercise is the most successful non-pharmacological method for reducing the risk of cardiovascular diseases [7].

Hypertension and abnormal cholesterol levels are two important conditions within the cluster of changes responsible for metabolic syndrome [8], a term conventionally used to refer to an increased risk of cardiovascular disease [9]. Although there is controversy regarding elevated serum lipid levels as a risk factor for mortality in the older population [10,11], there is no doubt that the combination of these conditions (i.e., hypertension and elevated triglycerides/reduced high-density lipoprotein) may be related to an increased risk of cardiovascular disease [12,13]. Considering the fundamental role of physical exercise in healthy aging, a multicomponent training program (MTP) emerges as a possibility of exercise intervention that consists of combining various types of training such as strength training, aerobic training, balance, coordination, and flexibility training [14]. In this context, several studies have already shown the benefits of an MTP on hemodynamic parameters [15,16] and functional capacity [17], as well as improvements in the lipid profile of older women [18,19]. Therefore, this type of training is recommended as a strategy to improve cardiovascular health [20] and physical functioning [21,22], and is advised for older adults in the long term [23,24]. Nonetheless, these individuals are often exposed to physical and behavioral factors [25], including seasonality [26], that result in greater risks of discontinuing an exercise habit. It is, therefore, crucial to further clarify both the short- and long-term effects and retention levels following the interruption of an exercise regimen [27,28]. When a person stops performing physical exercise regularly or there is a training stimulus of insufficient magnitude [29], they may suffer from the partial or total loss of positive adaptations induced by training, which is commonly called detraining (DT) [30]. In this line, studies involving detraining periods after an MTP have demonstrated that a few weeks (i.e., 4 to 8 weeks) of physical training interruption are enough to cause a significant decline in the functional capacity [31] and lipid profile [32] of older people. Regarding hemodynamic parameters, Leitão et al. [22] also observed a significant decrease in systolic blood pressure during the first 3 months of detraining after an MTP in older women.

In recent decades, some studies have examined the effects of strength and resistance retraining in active older people through measures related to strength, power, and muscle mass [33,34,35], in addition to postural control [36], but only a few researches focused on retraining based on an MTP [37]. The optimal retraining period for a trained older person to recover their FC after DT is an important aspect [37], specifically because older people usually interrupt their physical training programs when they spend summer vacations with their family [36], a period that may be prolonged for 2 to 3 months of DT [38]. Understanding the adaptations that come from retraining with an MTP can lead older adults to not stop training until they recover their functional capacity, lipid, and hemodynamic profile. Thus, our aim was to analyze the effect of 8 weeks of retraining with an MTP on the functional capacity and lipid and hemodynamic profile after 12 weeks of detraining in prehypertensive older women who had practiced 36 weeks of an MTP previously.

## 2. Materials and Methods

### 2.1. Sample

Thirty-eight prehypertensive older women (age range: 65–71 years) were recruited for the present study and randomly assigned into the experimental group (EG, *n* = 23, age: 68.3 ± 2.8 years and 1.61 ± 0.44 m, mean ± SD) or the control group (CG, *n* = 15, 66.3 ± 3.2 years; 1.59 ± 0.37 m, mean ± SD). The EG performed 36 weeks of the MTP followed by 12 weeks of detraining and 8 weeks of retraining. The CG maintained its daily routine and did not participate in structured physical training. Eleven participants did not complete the study, two in the EG (booth due to illness) and nine in the CG (two due to illness, and seven did not attend all assessments). Exclusion criteria were older women who did not have prehypertension, high total cholesterol (TC) and triglyceride (TG) levels, musculoskeletal disorders that could be aggravated by physical exercise, heart problems, and/or any medical contraindication.

### 2.2. Procedures

Throughout the study follow-up, participants’ detraining and retraining responses were assessed at baseline, after 24 and 36 weeks of the MTP, after 4 and 12 weeks of detraining, and after 8 weeks of MTP retraining. Functional capacity (FC) was assessed using 6 min walk (6-MWT) with heart rate peak (HRPeak), 30 s chair stand test (30-CS), handgrip strength (HGS), and 8-foot up-and-go (8-FUG). Metabolic parameters were also assessed, namely TC, GL, and TG, and hemodynamic profile (SBP, mmHg; and DBP, mmHg). The CG was measured at the same time points, without performing the MTP training period. All subjects provided their written consent after being informed about all study procedures, including the risks and benefits associated with it, as well as the possibility of withdrawing from it at any time. The study protocol was previously approved by the local institutional ethics committee (protocol number 2,887,652), according to the Declaration of Helsinki. All the variables in this study were assessed under the same environmental conditions on mornings between 10 a.m. and 12 p.m. at 22 to 24 degrees and 55–65% humidity by the same study team. Furthermore, all participants were instructed not to drink coffee, drink alcohol, smoke, or exercise in any way.

#### 2.2.1. Training Intervention

EG subjects participated in a supervised MTP containing 3 weekly sessions, totaling 108 sessions (compliance rate > 80%). Each training session lasted ~45 min, including a dynamic warm-up and cool-down phase. The protocol consisted of upper- and lower-limb exercises to promote aerobic and muscle endurance training. Each session was prescribed [21,34] and monitored by a specialist in sports exercise for older adults. The structure of the sessions started with a warming-up phase (i.e., low-intensity walking, calisthenics, and stretching exercises) and closed with a cool-down period (i.e., static and dynamic stretching techniques) using music to promote social and well-being to all participants. Intensity during the aerobic exercise and the training phase was kept within 2–3 points, according to the perceived exertion scale (RPE), and increased gradually to 4–5 points after the first four weeks of the MTP. In the muscular endurance training phase, all total body exercises were performed with 20 to 30 s of rest between sets in a circuit with one’s body weight (e.g., arm raise, heel to toe walking, back leg raises, air squat, obstacle overpass), adjacent to coordination exercises with agility and mobility tasks. The intensity progression was gradual, with the first month of training used to create adequate familiarization and adaptation to the exercises. Also, a resting period of 1 to 3 min was given between exercise prescriptions.

#### 2.2.2. Detraining Period (DT) and Retraining (RT)

To check the responses to DT, all EG and CG participants were assessed after 12 weeks of the MTP. All participants were asked during DT to resume their day-to-day routine and keep up their nutritional standards while avoiding any form of physical exercise on a regular basis. To monitor these questions, according to the researchers who applied the MTP, all participants were systematically contacted by telephone. This was performed to guarantee that they were complying with the DT conditions. After DT, the EG performed 8 weeks of retraining with the same training methodology as the 36-week exercise intervention.

### 2.3. Health Profiles and FC Assessments

The anthropometric assessment included body mass (kg) and stature (m). Body mass (kg) and body fat percentage (%BF) were determined using a portable scale (OMRON BF 303, Matsusaka, Japan) with a precision of 0.1 kg, and height was determined with a portable stadiometer (Seca Body meter^®^, model 208, Berlin, Germany) with a precision of 0.1 cm. BMI was calculated according to the formula (BMI = body mass/stature^2^). The standardized procedures described in previous studies were followed [39]. The measurements were taken in the standing position with minimal clothing, according to the principle of bioelectrical impedance analysis. The lipid profile (TC, GL, and TG) was assessed using a Cobas Accutrend Plus (Roche Diagnostics GmbH, Mannheim, Germany) with blood collected from the distal palmar phalanx of the third finger of the right hand using the Accu-Chek Softclix^®^ Pro lancing pen. The participants underwent a 12 h overnight fast and were off all medication for the hemodynamic profile. BP was measured after resting for 15 min in the supine position using a digital sphygmomanometer for older women (Omron HEM-907, Omron Healthcare, Europe B.V., Matsusaka, Japan). To analyze the FC of older women, we used Rikli and Jones’ [40] method and performed the following tests: 30-CS, 8-FUG, and 6-MWT. Also, HRPeak during 6-MWT was registered with the Garmin HRM-RUN strap. This battery of tests assesses the functional strength of the lower limbs, transition movements, balance, and risk of falling. After a 10 min warm-up with physical exercises, the participant was instructed to perform each test in the circuit. HGS was also measured according to Benton et al. [41] with Jamar 5030 J 1 (Jamar Technologies, Horsham, PA, USA) [42]. The best value of three records was used for this test.

### 2.4. Statistical Analysis

Analysis of the data was carried out using SPSS 19.0 for Windows (SPSS Inc., Chicago, IL, USA). Descriptive procedures of central tendency and dispersion to characterize the values of the variables were used, and we checked the normality of our sample using the Shapiro–Wilk test. The mean ± standard deviation was used for all data, with a two-tailed *p*-value ≤ 0.05 required for significance. A repeated measures ANOVA was used to compare all assessment points, followed by Bonferroni’s post hoc test. Sample size calculation with G*Power (0.3 effect size, a power of 83%, and significance *p* < 0.05) resulted in twenty-one older women for each group. The sphericity assumption was verified by Mauchly’s test. We also used Cohen’s d-effect size (ES): 0.2 or less small ES; about 0.5 or moderate ES; and 0.8 or more ES. Delta percentage (∆%) was used via ∆% = [(posttest score − pretest score)/pretest score] × 100.

## 3. Results

The EG’s responses to training, DT, and RT were interesting. In the first 24 weeks of exercise intervention, only SBP did not improve among all study variables. After 36 weeks of the MTP, positive changes occurred in BF% (ES: 1.39; *p* < 0.00; CI: 35.93–37.30), HGs (ES: 1.05; *p* < 0.00; CI: 18.20–19.34), 6MWT (ES: 2.27; *p* < 0.00; CI: 641.674.33), 8-FUG (ES: 1.70; *p* < 0.00; CI: 5.22–5.50), 30-CS (ES: 1.87; *p* < 0.00; CI: 21.60–23.40), TC (ES: 2.59; *p* < 0.00; CI: 194.54–201.54), TG (ES: 2.49; *p* < 0.00; CI: 207.27–212.07), and DBP (ES: 1.16; *p* < 0.00; CI: 81.08–83.51). Relating to the first 24 weeks and 36 weeks of the MTP, only GL, DBP, and BF% stagnated (Table 1).

With detraining (Table 2), we observed that only TC, GL, and 30-CS retained the benefits of the MTP after 4 weeks of DT. The blood pressure response to 12 weeks of DT resulted in an increase in SBP (ES: 0.42; *p* < 0.00; CI: 132.24–139.84) and DBP (ES: 1.02; *p* < 0.00; CI: 83.95–85.89). Compared to baseline values, only HGS (ES: 0.61; *p* < 0.01; CI: 17.60–18.71), 6-MWT (ES: 1.23; *p* < 0.01; CI: 601.72–633.11), 30-CS (ES: 1.36, *p* < 0.01; CI: 20.50–22.33), and TC (ES: 0.89; *p* < 0.01; CI: 207.69–215.39) improved.

The response to 8 weeks of RT (Table 2) was positive in all variables (*p* < 0.05), which allowed for the recovery of the benefits lost with 12 weeks of DT. In addition, it promoted even more significant improvements in HGS (ES: 0.51; *p* < 0.01; CI: 18.88–20.09), 30-CS (ES: 0.38; *p* < 0.04; CI: 22.52–23.90), and BP(SBP: ES: 0.36; *p* < 0.01; CI: 125.14–132.36; DBP: ES: 0.44; *p* < 0.02; CI: 80.04–82.13) compared to 36 weeks of MTP.

## 4. Discussion

Thirty-six-weeks of MTP resulted in significant improvements (*p* < 0.05) in lipid profile (i.e., TC and TG), hemodynamic profile (i.e., SBP and DBP), and functional capacity (i.e., 30-CS, 8-FUG, 6-MWT, and handgrip). Furthermore, other parameters, such as BF% and HRpeak, also improved significantly (*p* < 0.05). Longitudinal MTP interventions for 8 [18,43] to 9 months [44] have been shown to be an important physical exercise strategy to improve lipid profile through reductions in TC and TG in older women and are in line with our findings. The improvements in these parameters may be partially attributed to the combination of aerobic training and strength training [45] adopted in the MTP protocol performed in our study. In this regard, some reviews involving aerobic training [46] and strength training [47] have already demonstrated the benefits of both types of exercise in reducing the serum levels of these lipid biomarkers. Although the mechanisms relating to the effect of physical exercise on the lipid profile are still unclear [48], it appears that physical exercise may enhance the ability of skeletal muscle to use lipids to the detriment of glycogen [49], and thus reduce blood lipid levels in the long term. The underlying mechanisms responsible for these physiological changes may include increases in the lecithincholesterol acyltrans enzyme, which is responsible for the transfer of esters to high-density lipoprotein cholesterol [50], as well as significant increases in the level of lipase protein activity [51]. In addition, although Leite et al. [52] have verified that 12 weeks of strength training was better than the MTP in improving BF% through gains in lean mass, it is possible that these improvements may also happen when the MTP is applied in longitudinal interventions, such as in our study.

Just like the lipid profile, improvements in the hemodynamic profile may also be partly explained by the combination of exercises present in the MTP, which reap specific benefits from aerobic training and strength training [53]. This claim is confirmed by a recent systematic review with a meta-analysis, which found that the combination of both exercise modalities may significantly reduce SBP and DBP in postmenopausal women [54]. Additionally, other investigations that used the MTP in their physical exercise intervention were also in accord with our study [22,53,55], demonstrating significant reductions in blood pressure. Furthermore, similar to our study, Cornelissen et al. [7] reported that reductions in SBP and DBP were attributed to cardiorespiratory improvements. While the physiological mechanisms related to the reduction in BP have not been completely clarified, neurohormonal [56,57,58], structural, and functional vascular adaptations have been proposed [57]. It is proposed that physical exercise may have the ability to reduce sympathetic nervous activity and the subsequent release of norepinephrine [59], decreasing endothelin-1 levels [60] and increasing nitric oxide production [61], thus reducing vasoconstriction and peripheral vascular resistance [62]. These physiological adaptations together highlight the role of physical exercise as a non-pharmacological strategy for reducing blood pressure [15].

Aging consists of a natural physiological process that involves the gradual deterioration of functional capacity [17], specifically the neuromuscular and cardiorespiratory systems, which together largely determine an individual’s physical fitness levels [63]. The 30-CS is a test commonly used to measure the functional strength of the lower limbs in older people [64], in which the individual performs the sit–stand–sit movement without using their arms as many times as possible in a chair with a standardized height for 30 seconds [65]. In this variable, our results demonstrated a significant increase in the number of repetitions (*p* < 0.05), which is in line with previous studies after a period of the MTP [66,67], including clinical relevance for this population [68]. The 8-FUG test is widely used to assess functional capacity [69] through agility and dynamic balance in older people [70]. In this regard, our results showed significant improvements that are compatible with other studies that analyzed the MTP effects over a period of 12 weeks to 6 months in older people [71,72]. 6-MWT is a simple and inexpensive assessment tool frequently used to examine physical endurance in older people [73]. It measures the maximal distance covered by a subject who performs an individualized 6 min walk [74]. Regarding these parameters, our results are in agreement with previous studies that found an increase in the distance covered during this test after a period of the MTP in older people [75,76]. Sousa et al. [77] showed that combining aerobic training and strength training was more effective in improving 6-MWT performance than aerobic training performed alone in older men. The handgrip test is the most used muscle strength tool in clinical and scientific settings [78], standing out as an important health indicator in older people [79]. In agreement with a previous study involving the MTP [80], our results showed that this training form was effective in improving general muscle strength measured by the handgrip test in older people.

Three months of detraining caused a significant decline (*p* < 0.05) in functional capacity (i.e., 30-CS, 8-FUG, 6-MWT, and HGs), lipid (i.e., TC and TG), and hemodynamic profiles (i.e., SBP and DBP), in addition to BF%, which were similar to those presented in studies that analyzed this same period of detraining after the MTP in older women [22,30,81]. Regarding the lipid and hemodynamic profile, Leitão et al. [22,30] also showed that detraining for 3 months after a period of the MTP increased serum TC and TG levels, in addition to increasing SBP and DBP, in older women. These associated results are particularly worrying due to the strong association between high blood pressure, high total cholesterol and triglyceride levels, and the increased risk of cardiovascular death, including in the older population, which is demonstrated in robust cohort studies [82,83,84]. In general, studies that investigated aspects related to the loss of functional capacity after a period of 3 months of detraining after the MTP also observed a significant decline in the 30-CS, 8-FUG, 6-MWT [22,30,81], similar to our study. In addition, a significant decrease (*p* < 0.05) in HRpeak during 6MWT and HGs at the end of 3 months of post-MTB detraining was found. The decreases in HRpeak values as a direct outcome of age and physical inactivity might be undesirable because of their connection with an overall increased stroke risk [85]. Similarly, decreases in handgrip are also concerning, as declines in performance on this test are related to morbidity and mortality from all causes of cardiometabolic multimorbidity [86]. In this way, the detraining process leads the older person to a condition of loss of functional and physical capacity and has a negative impact on their capacity to carry out simple activities of day-to-day living, deteriorating a person´s quality of life [17] and, consequently, their autonomy, causing dependence.

Detraining–retraining is a cycle constantly faced by physically active older people and has been recently studied in physical training programs that involve strength training [36] and the MTP [37]. Lee et al. [37] observed that functional fitness in parameters related to the lower limbs (i.e., 30-CS) required 3 months to recover the post-training condition. For the parameters of aerobic endurance (i.e., 2 min step) and dynamic balance (i.e., 8-FUG), 9 months was necessary to gradually recover the post-training condition. Similarly, when we compared the end of the 8 weeks of retraining with the 36 weeks post-MTP, our results showed no statistical difference in aerobic endurance (i.e., 6-MWT) and dynamic balance (i.e., 8-FUG), indicating that these 8-week interventions were enough to recover the benefits lost with DT. Unlike Lee et al. [37], lower limb strength (i.e., 30-CS), HGs, SBP, and DBP improved with this short period of training compared to the levels after 36 weeks of the MTP. While all variables included in the present study improved significantly after 8 weeks of retraining compared to the end of 3 months of detraining, the delayed recovery of functional capacity levels regarding post-MTP may be partially explained by the enhanced effect of detraining, which leads to a return of muscle capillarization to the baseline value, presenting a decline of 25% to 45% in the activities of oxidative enzymes, which results in a reduction in mitochondrial ATP production in skeletal muscle [87]. Additionally, although some level of muscular strength may be maintained with detraining due to long-lasting neural adaptations [88], there is a tendency for a decrease in the size of type II fibers after detraining [89], which significantly impacts the functional capacity of older people. As such, the information presented in this article can help physical education professionals encourage older people not to interrupt the retraining process until functional capacity and lipid profile are completely restored concerning post-MTP levels. Furthermore, considering that some of these parameters can be recovered in the short term, new research should investigate minimum training strategies to be applied during the vacation period of older people and, thus, minimize the loss of functional capacity and accelerate the recovery process of the physical capacity achieved after the training period.

## 5. Conclusions

The response to the short-term period of 8 weeks of the MTP by the prehypertensive older women in this study enabled them to recover from a long-time period of DT, which started to affect them negatively after only four weeks. The systematic practice of 36 weeks of exercise contributed to a greater regulation of BP since 24 weeks of the MTP proved to not be enough to promote positive effects on SBP. In order to extrapolate the results of this sample, studies with a larger number of participants must be conducted.

## Figures and Tables

**Table 1 healthcare-12-00271-t001:** Follow-up of the MTP.

		Baseline			CI			24 Weeks of the MTP			CI			36 Weeks of the MTP			CI		
EG	BF% (%)	38.74	±	1.33	38.11	−	39.24	36.61	±	1.22 ^a^	36.04	−	37.07	36.68	±	1.62 ^b^	35.93	−	37.30
	HGS (kg)	17.36	±	1.43	16.70	−	17.91	18.34	±	1.51 ^a^	17.64	−	18.91	18.82	±	1.34 ^b^	18.20	−	19.34
	HRPeak (bpm)	129.81	±	2.23	128.85	−	130.73	135.57	±	2.36 ^a^	134.55	−	136.54	138.62	±	2.57 ^b^	137.50	−	139.67
	6MWT (m)	573.57	±	34.21	559.14	−	588.03	626.19	±	36.20 ^a^	610.88	−	641.45	657.57	±	39.46 ^b^	641.00	−	674.33
	8-FUG (s)	5.93	±	0.33	5.79	−	6.07	5.42	±	0.35 ^a^	5.28	−	5.57	5.36	±	0.34 ^b^	5.22	−	5.50
	30-CS (rep)	18.43	±	2.24	17.47	−	19.36	21.43	±	2.19 ^a^	20.49	−	22.34	22.52	±	2.13 ^b^	21.60	−	23.40
	TC (dL/ml)	219.24	±	8.10	215.70	−	222.55	203.00	±	8.52 ^a^	199.44	−	206.64	198.04	±	8.29 ^b^	194.54	−	201.54
	TG (mg/dL)	225.38	±	6.78	222.34	−	228.07	214.62	±	6.09 ^a^	211.84	−	216.99	209.81	±	5.68 ^b^	207.27	−	212.07
	GL (mg/dL)	94.71	±	3.91	92.97	−	96.28	91.62	±	3.96 ^a^	89.87	−	93.21	90.19	±	3.68 ^b^	88.61	−	91.72
	SBP (mmHg)	140.33	±	2.32	139.36	−	141.31	134.81	±	10.17	130.41	−	139.00	132.10	±	9.91 ^b^	127.86	−	136.23
	DBP (mmHg)	85.67	±	2.93	84.39	−	86.86	82.24	±	3.37 ^a^	80.83	−	83.67	82.29	±	2.88 ^b^	81.08	−	83.51
CG	BF% (%)	38.86	±	1.24	38.14	−	39.58	39.09	±	1.43	38.43	−	39.75	39.05	±	1.14	38.23	−	39.87
	HGS (kg)	17.34	±	1.36	16.55	−	18.12	17.37	±	0.73	16.95	−	17.80	17.61	±	0.71	17.20	−	18.02
	HR Peak (bpm)	130.79	±	2.64	129.26	−	132.31	131.00	±	1.47	130.15	−	131.85	131.36	±	1.45	130.52	−	132.19
	6-MWT (m)	577.86	±	39.84	554.85	−	600.86	577.50	±	34.18	557.77	−	597.24	575.71	±	37.82	553.88	−	597.55
	8-FUG (s)	5.83	±	0.17	5.73	−	5.93	5.85	±	0.15	5.77	−	5.94	5.87	±	0.15	5.79	−	5.96
	30-CS (rep)	16.50	±	1.09	15.87	−	17.13	16.71	±	1.27	15.98	−	17.45	17.07	±	1.77	16.05	−	18.10
	TC (dL/ml)	205.57	±	12.48	198.37	−	212.78	205.21	±	13.28	197.55	−	212.88	202.64	±	11.52	195.99	−	209.29
	TG (mg/dL)	182.29	±	19.26	171.17	−	193.40	180.29	±	18.01	169.89	−	190.69	180.79	±	13.99	172.71	−	188.86
	GL (mg/dL)	87.36	±	3.82	85.15	−	89.56	87.29	±	3.56	85.23	−	89.34	86.64	±	4.18	84.23	−	89.06
	SBP (mmHg)	144.64	±	2.65	143.11	−	146.17	144.00	±	2.18	142.74	−	145.26	144.21	±	1.37	143.42	−	145.01
	DBP (mmHg)	86.86	±	1.92	85.75	−	87.96	86.57	±	2.14	85.34	−	87.81	86.79	±	1.72	85.79	−	87.78

CI: confidence interval; BF%: body fat percentage; HGS: handgrip strength; HRPeak: heart rate peak in a 6 min walk test; 6-MWT: 6 min walk test; 8-FUG: 8-foot up-and-go; 30-CS: 30 s chair stand; MTP: multicomponent training program; DT: detraining; TC: total cholesterol; TGs: triglycerides; GL: glucose; SBP: systolic blood pressure; DBP: diastolic blood pressure. ^a^ Baseline vs. 24 weeks of the MTP, *p* < 0.05; ^b^ baseline vs. 36 weeks of the MTP, *p* < 0.05.

**Table 2 healthcare-12-00271-t002:** Follow-up of DT and RT.

		4-Week DT			CI			12 Weeks of DT			CI			8 Weeks of RT			CI		
EG	BF% (%)	37.64	±	1.87 ^a^	36.77	−	38.35	38.94	±	1.94 ^b^	38.04	−	39.68	37.57	±	1.70 ^c^	36.78	−	38.22
	HGS (kg)	18.65	±	1.35	18.03	−	19.17	18.20	±	1.32 ^b^	17.60	−	18.71	19.53	±	1.42 ^c^	18.88	−	20.09
	HRPeak (bpm)	136.00	±	3.14 ^a^	134.63	−	137.29	129.76	±	3.44 ^b^	128.30	−	131.20	134.67	±	4.37 ^c^	132.78	−	136.47
	6-MWT (m)	641.14	±	36.68 ^a^	625.64	−	656.61	617.43	±	37.16 ^b^	601.72	−	633.11	668.10	±	37.84 ^c^	652.77	−	684.73
	8-FUG (s)	5.55	±	0.35 ^a^	5.40	−	5.70	5.84	±	0.35 ^b^	5.69	−	5.98	5.37	±	0.25 ^c^	5.26	−	5.47
	30-CS (rep)	22.33	±	2.10	21.41	−	23.18	21.43	±	2.17 ^b^	20.50	−	22.33	23.24	±	1.64 ^c^	22.52	−	23.90
	TC (dL/ml)	199.33	±	9.04	195.56	−	203.19	211.57	±	9.12 ^b^	207.69	−	215.39	199.81	±	9.16 ^c^	195.93	−	203.66
	TG (mg/dL)	215.10	±	5.84 ^a^	212.45	−	217.38	222.29	±	7.24 ^b^	218.94	−	225.06	214.14	±	7.33 ^c^	210.82	−	217.01
	GL (mg/dL)	90.95	±	3.83	89.26	−	92.49	93.24	±	3.92 ^b^	91.51	−	94.82	91.14	±	3.90 ^c^	89.44	−	92.73
	SBP (mmHg)	134.57	±	9.17 ^a^	130.63	−	138.37	136.05	±	9.00 ^b^	132.24	−	139.84	128.76	±	8.54 ^c^	125.14	−	132.36
	DBP (mmHg)	83.90	±	2.49 ^a^	82.82	−	84.93	84.95	±	2.30 ^b^	83.95	−	85.89	81.10	±	2.48 ^c^	80.04	−	82.13
CG	BF% (%)	39.06	±	1.34	38.28	−	39.83	39.09	±	1.01	38.50	−	39.67	39.09	±	0.95	38.55	−	39.64
	HGS (kg)	17.66	±	0.93	17.13	−	18.20	17.84	±	0.63	17.48	−	18.21	17.73	±	0.50	17.44	−	18.02
	HRPeak (bpm)	132.07	±	2.46	130.65	−	133.49	131.79	±	2.58	130.30	−	133.27	131.14	±	1.96	130.01	−	132.27
	6-MWT (m)	578.93	±	41.70	554.85	−	603.01	580.14	±	18.55	569.43	−	590.86	581.64	±	14.62	573.20	−	590.08
	8-FUG (s)	5.81	±	0.28	5.68	−	5.92	5.81	±	0.18	5.79	−	5.95	5.85	±	0.18	5.75	−	5.96
	30-CS (rep)	16.64	±	1.98	15.50	−	17.79	16.43	±	1.16	15.76	−	17.10	16.36	±	1.08	15.73	−	16.98
	TC (dL/ml)	204.57	±	11.04	198.20	−	210.94	201.21	±	4.25	198.76	−	203.67	200.57	±	4.48	197.98	−	203.16
	TG (mg/dL)	183.36	±	15.70	174.29	−	192.42	180.64	±	8.48	175.75	−	185.54	180.71	±	8.05	176.07	−	185.36
	GL (mg/dL)	87.64	±	4.67	84.95	−	90.34	88.14	±	4.47	85.56	−	90.72	88.57	±	3.82	86.37	−	90.78
	SBP (mmHg)	143.79	±	2.15	142.54	−	145.03	143.57	±	0.65	143.20	−	143.95	143.71	±	0.73	143.30	−	144.13
	DBP (mmHg)	87.14	±	2.03	85.97	−	88.32	86.64	±	0.74	86.21	−	87.07	86.71	±	0.61	86.36	−	87.07

CI: confidence interval; BF%: body fat percentage; HGS: handgrip strength; HRPeak: heart rate peak in a 6 min walk test; 6-MWT: 6 min walk test; 8-FUG: 8-foot up-and-go; 30-CS: 30 s chair stand; MTP: multicomponent training program; DT: detraining; TC: total cholesterol; TGs: triglycerides; GL: glicose; SBP: systolic blood pressure; DBP: diastolic blood pressure; ^a^ 36 weeks of the MTP vs. 4 weeks of DT, *p* < 0.05; ^b^ 36 weeks of the MTP vs. 12 weeks of DT, *p* < 0.05; ^c^ 12 weeks of DT vs. 8 weeks of RT, *p* < 0.05.

## Data Availability

The data presented in this study are available upon request from the corresponding author.

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
