# Peer review of "Detraining and Retraining Effects from a Multicomponent Training Program on the Functional Capacity and Health Profile of Physically Active Prehypertensive Older Women"

_healthcare, 2024, doi:10.3390/healthcare12020271_

Round 1

Reviewer 1 Report

Comments and Suggestions for Authors

The age range of the surveyed women was not provided. There is only information about the average age of women from the experimental and control groups.

There is no description in the 'Material and methods' section of the methods of measuring metabolic parameters.

The training process carried out among older women is described too generally.

The small population of surveyed women in the experimental and control groups does not allow generalization of the research results. Moreover, the experimental group (60%) and the control group (40%) are unequal, which makes it difficult to compare them. Therefore, the conclusions formulated in the last chapter are not justified. This needs a major rewrite.

Author Response

Dear Reviewer,

We are grateful for your consideration of this manuscript, and we also very much appreciate your suggestions, which have been very helpful in improving the manuscript. We also thank the reviewers for their careful reading of our text. All the comments we received on this study of all reviewers have been attended into account in improving the quality of the article, and we present our reply to each of them separately.

The age range of the surveyed women was not provided. There is only information about the average age of women from the experimental and control groups. There is no description in the 'Material and methods' section of the methods of measuring metabolic parameters.

A: We add age range (LINE 93) and description on 2.3 Health Profiles and FC Assessments (LINE-156-163)

The training process carried out among older women is described too generally.

A: We rewrite. (LINE-130-138)|

The small population of surveyed women in the experimental and control groups does not allow generalization of the research results. Moreover, the experimental group (60%) and the control group (40%) are unequal, which makes it difficult to compare them. Therefore, the conclusions formulated in the last chapter are not justified. This needs a major rewrite.

A: A sample size calculation was made with the G*Power program considering a 0.3 effect size, 83% of power and p < 0.05 of significance, a sample of 21 older women per group would be necessary. We add sentence in statistical analysis (LINE 178-179) and sentence about number of drop-outs in the study that justify why booth groups do not have the same participants. (LINE 98-100). We rewrite the conclusions (LINE-321-322).

Reviewer 2 Report

Comments and Suggestions for Authors

Dear Author/s,

I read your paper carefully. It aims to evaluate the effects of detraining (DT) and retraining (RT) on functional capacity and lipid and hemodynamic profile in prehypertensive older women who had practiced a multicomponent training program (MTP) for 9 months before 3 months detraining and 2 months of re-training. I think there are some important notes that you should attention to before publishing your article.

1- In the abstract section, the number of patients in the intervention group/ EG is stated as 25, while in the text it is mentioned as 23. What is the reason for this discrepancy?

2- The reason for the unequal number of patients in the intervention and control groups is also not mentioned. Please indicate. Since the control group had 15 patients and the intervention group had 23 patients, what is the reason for this unequal number of patients?

3- The criteria for patient selection, including inclusion and exclusion criteria, and patient withdrawal should be clearly stated separately. Did all individuals remain until the end of the study or did some leave the study? If they left, it should be written with the reason.

4- In line 106 of the text, the 6-month post-intervention evaluation time should also be mentioned, because later in the table, results, and discussion, information is provided about its changes. Therefore, you should add this assessment time to lines 105-107. Because you have evaluated the patients in terms of all parameters in the sixth month, as well as 9th month.

5- What is the abbreviation for the word IC at the top of the table? Do you mean confidence interval or CI? Please write below the table what this abbreviation stands for.

6- The results section of the article is written very briefly and everything is crammed into the tables. In my opinion, for a better understanding of the results by the reader, you should state the key points from the tables in the text in a comparative manner and with mention of statistics (with P value and confidence interval). In the text of the article, it has been mentioned several times that such and such change was significant, but the level of significance has not been stated. Please review all these cases in the text of the article and add P values to them.

and finally, the reporting of the results should be based on the CONSORT checklist. Especially the Methods section, which jumped straight to the intervention procedure, and did not transparently state the study design, location, time, characteristics of participants, etc.

Good Luck

Comments on the Quality of English Language

Dear Editor,

Thank you for sharing this article and giving me the chance to review it. It emphasizes the importance of consistent exercise to maintain health benefits in prehypertensive older women. I should say that despite reading the article several times, I did not understand exactly why there were 9 months of exercise, 3 months of cessation, and 2 more months of exercise again, what was the real reason for conducting such a process and what was the real necessity for it. Since my specialty is traditional medicine and I do not have much knowledge of statistics and methodology, I assume that since this article was not related to my field, I did not properly understand the rationale and necessity of conducting it. However, I request you to send this article to someone with expertise in "sports medicine" and an "epidemiologist/statistician" simultaneously to ensure that conducting this study added some valuable things to current knowledge and whether its method selection was correct or not.

Regards, 

Prof. Roshanak  Ghods

Author Response

Dear Reviewer,

We are grateful for your consideration of this manuscript, and we also very much appreciate your suggestions, which have been very helpful in improving the manuscript. We also thank the reviewers for their careful reading of our text. All the comments we received on this study of all reviewers have been attended into account in improving the quality of the article, and we present our reply to each of them separately.

1- In the abstract section, the number of patients in the intervention group/ EG is stated as 25, while in the text it is mentioned as 23. What is the reason for this discrepancy?

A: It was incorrect. We changed to 23. (LINE-26)

2- The reason for the unequal number of patients in the intervention and control groups is also not mentioned. Please indicate. Since the control group had 15 patients and the intervention group had 23 patients, what is the reason for this unequal number of patients?

A: We add sentence about drop-out to justify. "Eleven participants did not complete the study, two in EG (booth due to illness) and nine in CG (two due to illness and seven did not attended all assessments). " (LINE 98-100)

3- The criteria for patient selection, including inclusion and exclusion criteria, and patient withdrawal should be clearly stated separately. Did all individuals remain until the end of the study or did some leave the study? If they left, it should be written with the reason.

A: We rewrite exclusion criteria and add sentence about participants drop-out. (LINE-98-103)

4- In line 106 of the text, the 6-month post-intervention evaluation time should also be mentioned, because later in the table, results, and discussion, information is provided about its changes. Therefore, you should add this assessment time to lines 105-107. Because you have evaluated the patients in terms of all parameters in the sixth month, as well as 9th month.

A: We add. (LINE 105)

5- What is the abbreviation for the word IC at the top of the table? Do you mean confidence interval or CI? Please write below the table what this abbreviation stands for.

A: We add abbreviation description for Confidence Interval (CI).(LINE 203-204)

6- The results section of the article is written very briefly and everything is crammed into the tables. In my opinion, for a better understanding of the results by the reader, you should state the key points from the tables in the text in a comparative manner and with mention of statistics (with P value and confidence interval). In the text of the article, it has been mentioned several times that such and such change was significant, but the level of significance has not been stated. Please review all these cases in the text of the article and add P values to them. And finally, the reporting of the results should be based on the CONSORT checklist. Especially the Methods section, which jumped straight to the intervention procedure, and did not transparently state the study design, location, time, characteristics of participants, etc.

A: We perform CONSORT checklist and we add a subsection of Sample in methods section (LINE 92), and rewrite results section based on your suggestion (LINE 185-202). Also, we review all the cases in the text for p value (LINE 213, 216, 273, 285).

Reviewer 3 Report

Comments and Suggestions for Authors

Dear authors

The MS entitled “Detraining and retraining effects from a multicomponent training program on the functional capacity and lipid and hemodynamic profile of physically active prehypertensive older 4 women” was evaluated. The study reports responses of old age women bodies to react in case of detraining and retraining while monitoring their various parameters.  The study is interesting and was conducted systematically. The methods are according to standards while results were nicely presented. My suggestions have been provided.

1.       The title needs to be rearranged and rephrased. The repetition of training word is not suitable.

2.       Also, the abstract is not ok. Th first lines of background should be corrected. Rewrite the abstract.

3.       Line 27. Were these women 25 or 23 in numbers?

4.       66-73 sentences too long. Check for other longer sentences and make them short.

5.       89-92. Corrections needed in hypothesis, research gap and aims/objectives.

6.       191. B represent 9 months or 12 months? Clarify.

7.       Discuss the main results in a composite form while providing relevant graphs. No graphical representation.

8.       Corrections in references journal names (7 and 13 and others).

Comments on the Quality of English Language

English Language needs to be refined. 

Author Response

Dear Reviewer,

We are grateful for your consideration of this manuscript, and we also very much appreciate your suggestions, which have been very helpful in improving the manuscript. We also thank the reviewers for their careful reading of our text. All the comments we received on this study of all reviewers have been attended into account in improving the quality of the article, and we present our reply to each of them separately.

  1. The title needs to be rearranged and rephrased. The repetition of training word is not suitable.

A: We change to “Detraining and retraining effects from a multicomponent training program on the functional capacity and Health profile of physically active prehypertensive older women”

  1. Also, the abstract is not ok. Th first lines of background should be corrected. Rewrite the abstract.

A: We rewrite. (LINE: 22-24)

  1. Line 27. Were these women 25 or 23 in numbers?

A: We changed to 23. (LINE 26)

  1. 66-73 sentences too long. Check for other longer sentences and make them short.

A: We rewrite. (LINE 69-71)

  1. 89-92. Corrections needed in hypothesis, research gap and aims/objectives.

A: We rewrite sentence. (LINE 81-90)

  1. B represent 9 months or 12 months? Clarify.

A: We changed to 9-month. (LINE 206)

  1. Discuss the main results in a composite form while providing relevant graphs. No graphical representation.

A: We rewrite results with more detail in the text. Also, we add p values in all key points and in case of doubt, the reader can go back to the results section for more precise information (LINE 184-202).

  1. Corrections in references journal names (7 and 13 and others).

A: We changed. (In red on references section: LINE 344-571)

Round 2

Reviewer 2 Report

Comments and Suggestions for Authors

Dear authors, I have reviewed the revised file. Please take a look at the attached PDF file and make the necessary minimal revisions. Best of luck with your work.

Comments on the Quality of English Language

N/A

Author Response

Dear reviewer,

we attended your las suggestions.

thank you for your attention, 

best regards

Reviewer 3 Report

Comments and Suggestions for Authors

Dear authors. The corrections are ok. 

Comments on the Quality of English Language

English language is ok. 

Author Response

Dear reviewer,

thank you for your suggestions.

regards